# MUSCAT: a Multimodal mUSic Collection for Automatic Transcription of real recordings and image scores

Multimodal audio-image music transcription has been recently posed as a means of retrieving a digital score representation by leveraging the individual estimations from Automatic Music Transcription (AMT)—acoustic recordings—and Optical Music Recognition (OMR)—image scores—systems. Nevertheless, while proven to outperform single-modality recognition rates, this approach has been exclusively validated under controlled scenarios—monotimbral and monophonic synthetic data—mainly due to a lack of collections with symbolic score-level annotations for both recordings and graphical sheets. To promote research on this topic, this work presents the *Multimodal mUSic Collection for Automatic Transcription* (MUSCAT) assortment of acoustic recordings, image sheets, and their score-level annotations in several notation formats. This dataset comprises almost 80 hours of real recordings with varied instrumentation and polyphony degrees—from piano to orchestral music—1251 scanned sheets, and 880 symbolic scores from 37 composers, which may also be used in other tasks involving metadata such as instrument identification or composer recognition. A fragmented subset of this collection exclusively focused on acoustic data for score-level AMT—the *MUSic Collection for aUtomatic Transcription - fragmented Subset* (MUSCUTS) assortment—is also presented together with a baseline experimentation, concluding the need to foster research on this field with real recordings. Finally, a web-based service is also provided to increase the size of the collections collaboratively.

Additional Key Words and Phrases: Automatic Music Transcription, Optical Music Recognition, multimodal learning, data collection

ACM Reference Format:
. 2024. MUSCAT: a Multimodal mUSic Collection for Automatic Transcription of real recordings and image scores. 1, 1 (April 2024), 9 pages. https://doi.org/XXXXXXX.XXXXXXX

## 1 INTRODUCTION

Music transcription, the research field investigating how to computationally transcribe music sources into machine-readable formats, is deemed as one of the key processes within the Music Information Retrieval (MIR) area [30]. Depending on the nature of the source data, the MIR community depicts two different—yet related—research lines: (i) Automatic Music Transcription (AMT) when addressing acoustic music signals [4]; and (ii) Optical Music Recognition (OMR) when the source of data is a document [6]. However, while to some extent pursuing a similar goal, AMT and OMR proposals have been typically addressed resorting to domain-specific solutions and target notations (*e.g.*, piano-roll representations) due to the difficulty of defining a common transcription framework [2].

Author's address:

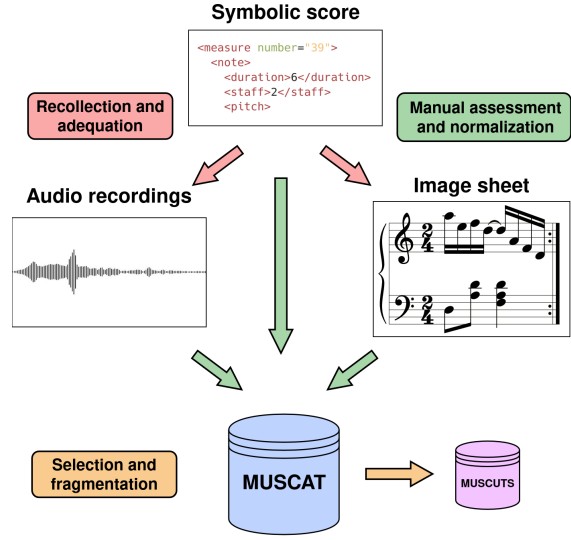

Fig. 1. Graphical description of the process carried out for creating the *Multimodal mUSic Collection for Automatic Transcription* (MUSCAT) collection, together with the subset of cut pieces *MUSic Collection for aUtomatic Transcription - fragmented Subset* (MUSCUTS). Taking a symbolic score as reference, the corresponding acoustic recording and image sheet are gathered from public sources. After that, these data are manually assessed and then discarded if the pieces do not represent the same work (or excerpt from it). Eventually, each modality undergoes a standardization process before being incorporated into the collection.

Nonetheless, recent proposals have modeled both AMT and OMR as *sequence labeling* tasks [16] resorting to *holistic* or *end-to-end* frameworks based on deep learning schemes. In these cases, the input datum—image or audio—is associated with a sequence of tokens as the output of the recognition system, without any input-output alignment requirement [7, 21]. Nevertheless, one of the main advantages of this formulation is that it allows for a common target representation in both transcription paradigms: a digital score encoded in a music-notation standard (*e.g.*, MusicXML or Kern).

Recently, music transcription has been framed within a multimodal formulation in which the acoustic recording and the sheet image constitute the individual modalities of a common target representation, a digital music score [2]. However, while proven to outperform single-modality recognition rates, this approach has been exclusively validated under controlled scenarios—monotimbral and monophonic synthetic data—due to the lack of existing corpora devised explicitly for this task.

This work presents the *Multimodal mUSic Collection for Automatic Transcription* (MUSCAT) collection of real audio recordings and image sheets together with notation-level digital scores. The assortment, which was manually compiled from different sources

Table 1. Summary of multimodal corpora depicting audio performances, image scores, and symbolic music notation. The presented MUSCAT and MUSCUTS collections are included for comparative purposes. Symbols R and S denote whether the modality is either real or synthetic, whereas superscript † specifies that this timbre was selected by the work in the specified task.

| Name | Audio recordings | Image scores | Symbolic encoding | Instrumentation | Polyphony | Duration (hours) | Freely available | Multimodal use case |
|---|---|---|---|---|---|---|---|---|
| Fremerey [14] | R | R | None | Piano, string | ✓ | 12 | ✗ | Sheet music-audio ident. |
| Notthingham | S | S | ABC | Piano† | ✓ | - | ✓ | Score following [10, 17] |
| Magaloff [13] | R | R | MusicXML | Piano | ✓ | 10.13 | ✗ | Score following [19] |
| Zeilinger [8] | R | R | MusicXML | Piano | ✓ | 3 | ✗ | Score following [19] |
| MSMD [11] | S | R | Lilypond | Piano | ✓ | 15 | ✓ | Score following [18] |
| MTD [34] | R | R | Sibelius, MusicXML | Voice, orchestra, piano, guitar strings, winds | ✓ | 4.96 | ✓ | Transcription retrieval, alignment, computational musicology |
| MeSA-13 [12] | R | R | MusicXML | Piano | ✓ | - | Audios | Score alignment |
| Primus [7] | S | R | Kern | Piano† | ✗ | 42.66 | ✓ | Transcription [2] |
| MUSCAT | R | R | Kern, Lilypond, MusicXML | Voice, orchestra, piano, guitar, organ violin, viola, cello | ✓ | 79.98 | ✓ | Transcription, score following style/composer identification |
| MUSCUTS | R | - | Kern, Lilypond, MusicXML | Voice, orchestra, piano, guitar, organ violin, viola, cello | ✓ | 22.19 | ✓ | Score-level audio transcription |

on the Internet and later curated to remove possible error sources, contains a set of real recordings comprising varied instrumentation—from piano to orchestral music—and different polyphony degrees that span for 80 hours together with 1251 scanned sheets and 880 symbolic scores from 37 composers. Besides, while MUSCAT is expected to act as a reference and benchmark collection for multimodal transcription, it may also be considered as raw data—*i.e.*, the different modalities are not aligned—to foster research on other tasks involving acoustic recordings and score data, such as sheet music-audio identification [14] or score following [17] after being manually aligned. Moreover, the provided metadata may be relevant to other classification-oriented tasks, such as composer recognition, difficulty estimation or style identification, among others.

One of the main challenges in multimodal music transcription is the remarkable performance difference between the involved modalities, as OMR schemes consistently outperform holistic score-level AMT proposals [2]. According to the related literature, this underdevelopment of the latter field is primarily due to a lack of data collections devised in this task [21, 24, 26]. In response, we introduce the *MUSic Collection for aUtomatic Transcription - fragmented Subset* (MUSCUTS) collection, a subset of MUSCAT particularly devised for holistic score-level AMT. This assortment consists of 2698 fragments of symbolic scores together with their corresponding recordings, adding up to a total of more than 22 hours of real audio. This dataset, which represents the largest collection of real-life recordings specifically devised for score-level AMT, is expected to promote the development of this field and eventually facilitate research on the multimodal transcription paradigm.

The rest of the work is structured as follows: Section 2 contextualizes MUSCAT among other existing data collections depicting audio and image information; after that, Section 3 introduces the details of the presented assortment as well as its MUSCUTS subsets, thoroughly describing their compilation and curation processes and detailing its main characteristics, together with some base results for the particular case of score-level AMT; then, Section 4 presents the facilitated web-based service to access the collections as well as to contribute to their growth; finally, Section 5 concludes the work and poses future research lines.

## 2 RELATED WORK

Attending to the multimodal nature of music [29]—*i.e.*, it can be represented as audio, in a symbolic representation, as a text or an image, among others—, a large number of MIR proposals have addressed the integration and leveraging of these individual sources of information [31]. Nonetheless, as aforementioned, no existing data collection has been specifically devised to address multimodal music transcription from audio and image due to the recentness of its formulation.

It must be noted that, while some data collections comprise the required modalities for this paradigm, none of them were devised for the target multimodal transcription paradigm, hence exhibiting some limitations for their use in this context. We now review some of the most relevant cases from the existing MIR literature and summarize their main features in Table 1, being the MUSCAT and MUSCUTS assortments also described for comparative purposes. For a detailed review of multimodal data collections for other MIR tasks, the reader is referred to the work by Christodoulou et al. [9].

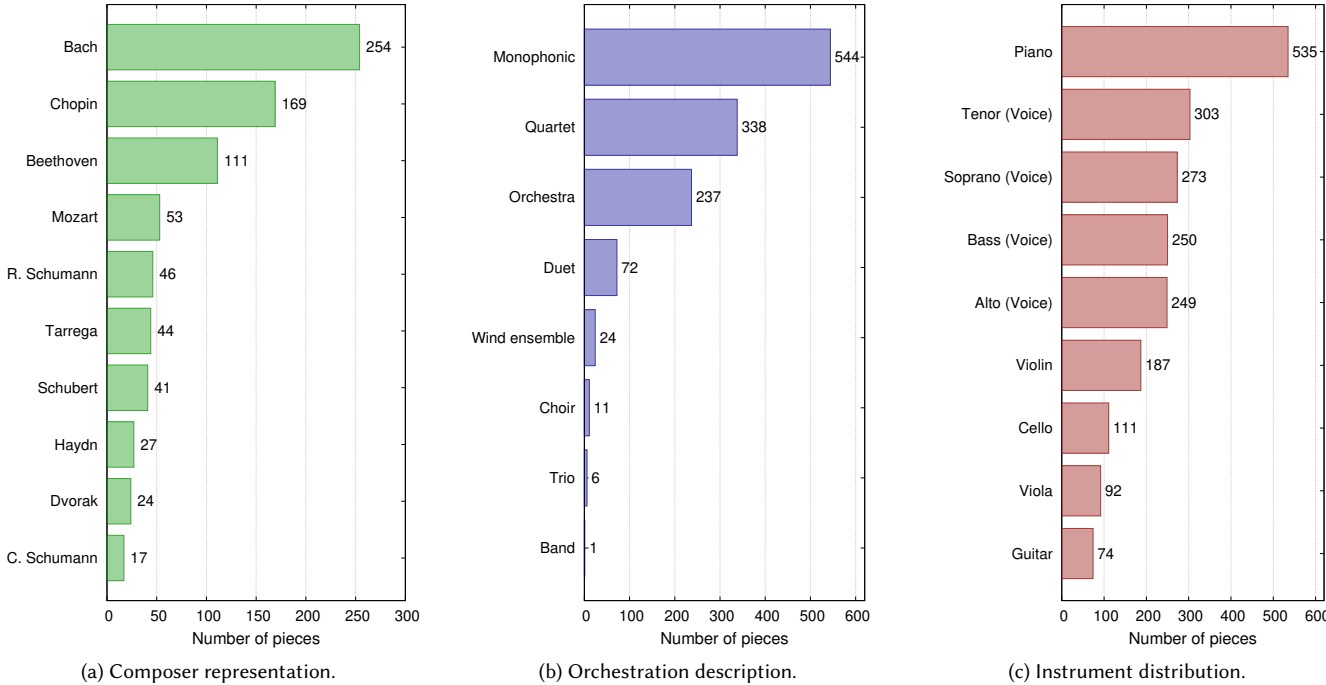

Fig. 2. General statistics of the MUSCAT collection.

A first example is the work by Fremerey et al. [14] for sheet music-audio identification, *i.e.*, finding the sections within an audio interpretation corresponding to a query consisting of a sequence of bars from a sheet music representation. Although the authors evaluated the proposal using a dataset collected explicitly for the work comprising real audio recordings and scanned sheet images, no symbolic transcription is provided.

The works by Dorfer et al. [10] and Henkel et al. [17] explored the use of the Notthingham[1] collection in the context of score following. Based on the symbolic codifications of the assortment, the authors synthesized the audio files and rendered the sheet scores. Nevertheless, the data is entirely synthetic, being hence of reduced application to evaluated multimodal transcription in real-world scenarios.

The Magaloff [13] and Zeilinger [8] collections of sheet scores and real recordings were also recently considered for score following [19]. Nevertheless, these sets exhibit a rather limited duration and only contain piano pieces. Similarly, the so-called *Multimodal Sheet Music Dataset* (MSMD) [11] dataset was also used for the score alignment tasks in [18], but the audio modality constitutes synthesized pieces from the MIDI files provided.

The *Musical Theme Dataset* (MTD) [34] was recently proposed to foster research on a variety of MIR tasks as, for instance, cross-modal music retrieval, music alignment, music transcription or computational musicology. This multimodal collection comprises 2067 themes in sheet music format with their corresponding audio

snippets of music recordings as well as symbolic music encodings. However, the limited size of the collection—roughly 5 hours of audio recordings—limits its application to the task at hand, at least for the training stage of deep learning methods.

To our best knowledge, the Printed Images of Music Staves (PrIMuS) corpus [7] is the only assortment that has been considered in multi-modal audio-image transcription frameworks [2]. While the size of the dataset and the availability of sheet scores make it suitable for this task, the unavailability of real recordings and the monophony of the pieces limit the reach of the conclusions obtained.

Alternatively, some authors have introduced different tools to facilitate—and, if possible, automate—the compilation of multimodal collections comprising audio recordings and sheet music. One of them is the *Audio-Score Meta-Dataset* (ASMD) framework [32] that, while originally devised for experimental reproducibility, was eventually extended to be used as a general data compilation suite. Similarly, the *Measure to Sound Annotator* (MeSA) [12] toolbox, a web application that interactively aligns sheet music and performances, was also introduced to facilitate multimodal data compilation. In addition, the authors also released the MeSA-13 set, comprising a set of 13 pieces to showcase the capabilities of the framework.

In this context, MUSCAT is proposed as a novel multimodal data collection for tasks involving audio recordings and sheet music. In contrast to the aforementioned corpora, MUSCAT exhibits the largest amount of real data for the two contemplated music modalities as well as the symbolic score-level transcription for a wide variety of instrumentation. Hence, this assortment is expected to

---

[1]ifdo.ca/~seymour/nottingham/nottingham.html

, Vol. 1, No. 1, Article . Publication date: April 2024.

foster research not only in multimodal audio-image transcription but also in topics such as the previously introduced score following and sheet music-audio identification topics, among others.

# 3 THE MUSCAT COLLECTION FOR MULTIMODAL AUDIO-IMAGE MUSIC TRANSCRIPTION

MUSCAT—acronym for *Multimodal mUSic Collection for Automatic Transcription*—constitutes the first dataset specifically devised for multimodal score-oriented transcription tasks. The collection comprises a set of real recordings with a total duration of almost 80 hours comprising a varied range of instruments—from piano to orchestral music—and polyphony degrees as well as 1251 scanned sheets and 880 symbolic scores from 37 different composers from the Common Practice Period. Figure 2 provides additional statistics related to the number of pieces per composer, the type of pieces in terms of the orchestration, and the instrument distribution.

The rest of the section describes the creation process of the corpus by detailing the individual collection and curation processes for the different modalities. In addition, the aforementioned MUSCUTS assortment is presented as a subset of the MUSCAT collection, and details related to its compilation are also provided. Finally, the last section delves into the limitations of two datasets and possible mechanisms to alleviate them in the future.

## 3.1 Collection and curation processes

Instead of relying on automatic web-scraping approaches, MUSCAT was manually compiled and curated to provide a reliable and accurate multimodal collection of music data. A dedicated professional musicologist carried out this process for 8 months, and our team subsequently reviewed the data.

Figure 1 graphically illustrates the devised process followed in the creation of MUSCAT. As it may be observed, the driving criterion is the availability of a symbolic score. Based on this, the musicologist manually collected the two data modalities—acoustic recording and image sheet—from publicly available resources. After that, the annotator corroborated that the different modalities actually represented the same musical piece—the *Manual assessment* stage—, being otherwise discarded. Eventually, if accepted, the modalities were post-processed—*Normalization* stage—to be distributed in a standard modality-wise format of the MUSCAT collection.

The rest of the section focuses on each data modality of the collection to accurately describe the *Adequation* and *Normalization* processes carried out in each case.

*3.1.1 Symbolic score.* Given that the main target of the presented assortment is transcription, the guiding element in our dataset creation pipeline is, as aforementioned, the symbolic music score. In this regard, while all works in the collection may not contain both the recording and image individual modalities, they all depict this digital score. Note that, in the absence of one of the modalities (or both), this digital representation may be used for synthesizing an execution of the piece—acoustic modality—and/or engraving the score as a printed document—image modality.

Regarding the notation, while many symbolic music representations exist, we exclusively resorted to scores in the MusicXML format [15] as it has been largely considered both for research and commercial applications. However, to align with previous score-level AMT and OMR efforts, we additionally provide all these pieces translated to the Humdrum **kern standard [20]—*e.g.*, used for AMT in [1, 3] and OMR in [28]—as well as the Lilypond format [25]—*e.g.*, used for AMT in [22, 23]. Figure 3 shows an example of these three symbolic representations provided in MUSCAT.

In terms of data sources, we mainly relied on three public Internet repositories commonly contemplated by MIR researchers: (i) MuseScore,[2] (ii) International Music Score Library Project (IMSLP)/Petrucci Music Library,[3] and (iii) MusOpen.[4] To complement these sources, additional works were gathered from particular websites, retrieving a total of 880 symbolic scores from the Common Practice Period. As commented, these files were manually verified by a musicologist, and the only automated process is converting the file from MusicXML to both the **kern and Lilypond formats.

*3.1.2 Audio recordings.* The second stage of the pipeline comprises compiling the acoustic performances corresponding to the symbolic scores gathered in the first phase. As commented, MUSCAT differs from most existing score-oriented transcription collections in that the acoustic data is gathered from real-performance recordings.

For that, we manually searched for real performances on the Internet with a suitable license that allowed its free distribution and use. These performances have been manually processed to remove spurious elements—*e.g.*, background noise from the start and/or end of the recording or sections only depicting hand-clapping from the audience—and, when possible, sliced into smaller units—*e.g.*, movement sections—to facilitate its processing. The gathered data comprises almost 80 hours of real-performance recordings. A more detailed description of the duration of the pieces is provided in Fig. 4.

Finally, the compiled recordings undergo a normalization process so that they all depict the same characteristics: stereo files with a sampling rate of 48kHz.

It must also be mentioned that, in some cases, there is more than one recording per musical piece, *i.e.*, the dataset contains different renditions of the same work. Such a fact is expected to be useful to further gain insights about the performance of transcription systems at hand by comparing the relation between the input elements—each of the renditions—and the target output score. Finally, note that this case of different performance recordings of the same piece is analogous to that of having different—but complementary—points of view of the same input datum, being hence suitable to be studied from a multi-view learning perspective [35].

*3.1.3 Image sheets.* Similarly to the audio recordings, the image sheets were manually compiled from different free sources on the Internet, taking the symbolic score from the first step as a reference. Note that no particular criteria in terms of printing style (*e.g.*, handwritten or typeset), color, or quality were considered not to bias the difficulty of the task. Figure 5 shows two excerpts of image sheets and a summary of the main characteristics of the data collected for this modality.

Once collected, all the images were manually edited so that the document only contained information about the music piece at hand.

---

[2]https://musescore.org/
[3]https://imslp.org/
[4]https://musopen.org/

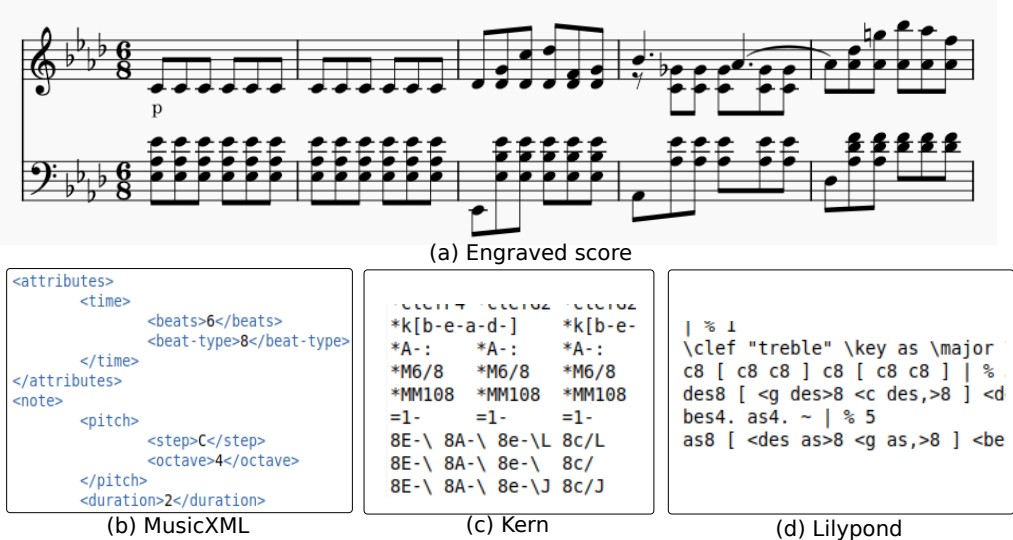

(a) Engraved score

(b) MusicXML

(c) Kern

(d) Lilypond

Fig. 3. Examples of the encoding formats provided in MUSCAT—Figs. (b) to (d)—as well as the reference rendered score with a music engraving system—Fig. (a).

That is, if at some point one page contained two or more scores from different pieces, the parts not belonging to the music work at hand were removed. No other processes involving resolution or image size changes were applied to these sheets.

### 3.2 The MUSCUTS assortment for score-level AMT

As aforementioned, due to the lack of real data for score-level AMT, this work also introduces the so-called *MUSic Collection for aUtomatic Transcription - fragmented Subset* (MUSCUTS) assortment. Such a collection constitutes a subset of the MUSCAT dataset particularly oriented towards score-level AMT tasks and, to our best knowledge, represents the largest collection of real data for score-level AMT.

It must be highlighted that, due to the limitations in the state-of-the-art score-level AMT systems regarding the temporal duration of the datum to be transcribed, additional annotation and curation processes were required to adapt the acoustic recordings in MUS-CAT to this scenario. Annotation and curation were carried out for three months by two professional musicologists using an in-house annotation tool, focused on diving the samples surpassing a threshold span of 30 seconds—together with the corresponding symbolic scores—into segments of, at most, this duration threshold. The original scores were cut at bar times to maintain the coherence of the gathered fragments.

Finally, while such temporal segmentation was particularly devised considering the limitations in current score-level AMT systems, this data preparation allows this assortment to be used for future AMT proposals not affected by this temporal-span limitation as the different segments may be joined considering the segmentation points provided.

#### 3.2.1 Baseline experimentation. As aforementioned, MUSCUTS seeks to fill a gap in the score-level AMT literature by providing a large

and challenging assortment of real acoustic recordings comprising a wide variety of instrumentation and polyphony degrees. In this sense, since most existing research proposals in the field have resorted to constrained and relatively simple experimental conditions, we consider that providing an initial assessment of this collection on the most competitive strategies for score-level AMT may provide insights about the maturity of the field.

For that, we have considered the proposal by Alfaro-Contreras et al. [1] that currently represents the state of the art in the field. The model is based on an encoder-decoder scheme in which the former stage extracts the adequate features for the transcription task considering a Convolutional Neural Network and the latter infers, on an auto-regressive basis, the score-level token associated with each time step. The proposal contemplates a Transformer architecture [33] to model the temporal dependencies of the sequence as well as a two-dimensional positional encoding mechanism adequately devised for this task. For the sake of conciseness, we omit the precise details of the neural scheme, which may be consulted in the reference work.

Regarding the input-output representation of the data, we consider the same experimental conditions as in [1]. In terms of the acoustic input, we employ a Short-Time Fourier Transform representation with log-spaced bins and log-scaled magnitude and a 2048-sample Hanning window with a hop size of 512 samples. Concerning the score representation, we adopt the Kern encoding and consider each symbol a single category.

About the training process of the model, we divided the MUS-CUTS assortment into three partitions—training, validation and test—, respectively corresponding to the 85%, 15% and 5% of the collection, being all excerpts of a piece assigned to the same set to avoid possible biases. As in the reference work, the model is trained for a maximum of 1000 epochs using the ADAM optimizer with a constant learning rate of $10^{-4}$ and a patience of 5 epochs.

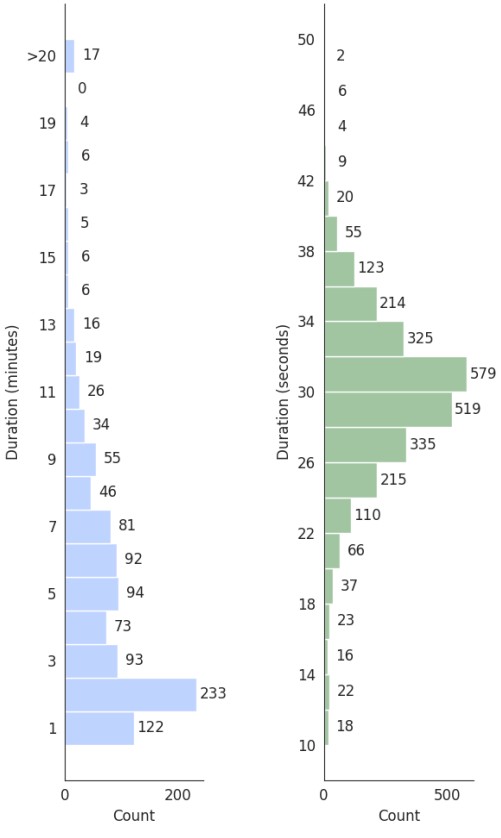

(a) Original recordings.  (b) Sliced acoustic recordings.

Fig. 4. Duration histogram of the audio recordings.

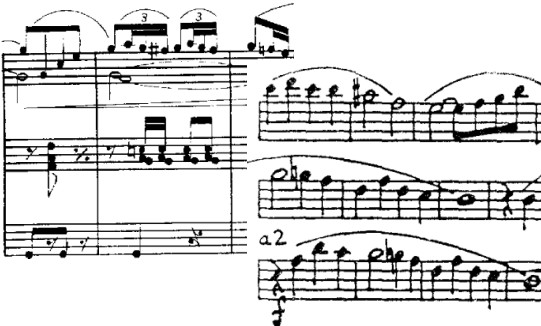

(a) Excerpts of two examples of scanned sheet scores in MUSCAT.

| Number of scores | Pages per score | Dimension ranges (pt.) | |
|---|---|---|---|
| | | Width | Height |
| 1251 | 9.1 | $[84 - 4982]$ | $[72 - 6850]$ |

(b) General statistics of the image sheet set in terms of the total number of scores, the number of pages per score, and the width and height ranges (in points) of the data.

Fig. 5. Overview of the image sheet modality.

Table 2. Comparative of the SER results obtained by reference works in notation-level AMT. A description of the type of data used in each case is also provided.

| Strategy | Data description | SER (%) |
|---|---|---|
| Alfaro-Contreras et al. [1] | Synthetic data with constrained polyphony | 5.3–15.7 |
| Martinez-Sevilla et al. [24] | Saxophone real recordings (monophonic data) | 22.9–55.7 |
| MUSCUTS | Real recordings with unconstrained polyphony | 75.5 |

Concerning the evaluation, we have resorted to the Symbol Error Rate (SER) due to its common use as a figure of merit in score-level transcription [2, 3, 21]. This metric reports the average number of elementary editing operations (insertions, modifications, or deletions) needed to match the predicted sequence with the reference one, normalized by the length of the latter.

Considering the presented experimental scheme, the results obtained are reported in Table 2. To illustrate the difficulty involved in transcribing real polyphonic audios, in this table we also report the range of SER values obtained in different data settings in reference works from the literature that evaluate simpler cases.

As it may be observed, the baseline results reported for the MUSCUTS collection (SER value of 75%) is considerably worse than that obtained by state-of-the-art methods in the field—SER values of, approximately, 5% to 15.7% in the work by Alfaro-Contreras et al. [1] with synthetic data or 22.9% to 55.7% in the case of Martinez-Sevilla et al. [24] of monophonic recordings—. However, far from being discouraging figures, the fact that the recognition model used in the work is based on that by Alfaro-Contreras et al. [1], proves MUSCUTS as a challenging assortment—real recordings with varied instrumentation and unconstrained polyphony degrees—to promote research in score-level AMT as the current state-of-the-art is not able to address such scenarios.

Finally, Figure 6 shows a graphical example of the results obtained with this transcription approach on a test sample of the MUSCUTS dataset.

### 3.3 Limitations

Since MUSCAT has been devised to tackle a very novel and particular research field—multimodal score-level transcription from acoustic recordings and image sheets—, the type of annotations may not be of direct application to all existing MIR tasks. In this regard, since the different modalities have not been aligned, specific applications

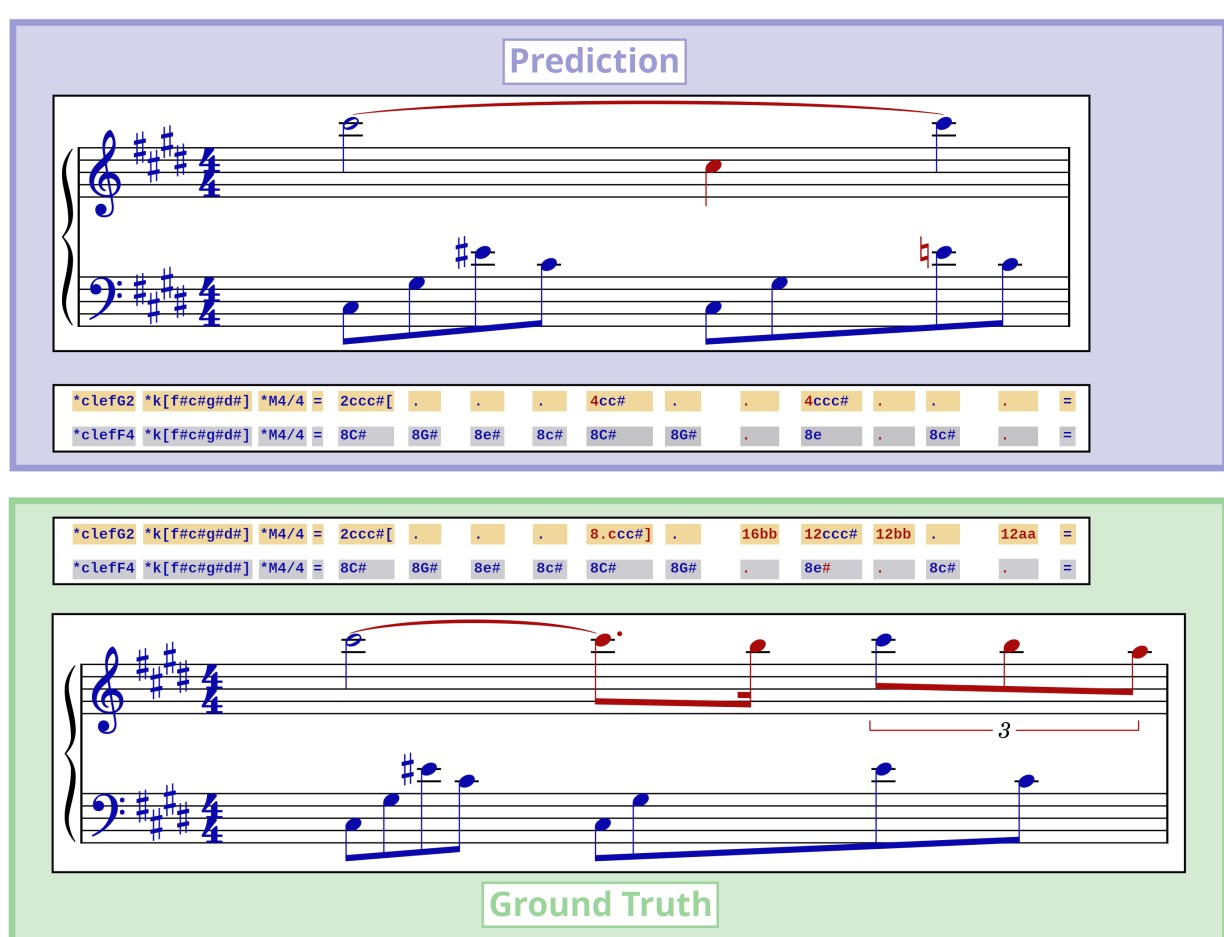

Fig. 6. Transcription measure example of the MUSCUTS test set. Concordances are colored in blue, while discordances are shown in red.

such as score following or sheet music-audio identification still require to undergo such an alignment stage—either manually or using assisted tools such as the one by Feffer et al. [12], for instance.

It must also be highlighted that since the data has been collected for a rather specific transcription formulation that is not directly compatible with classical event-based AMT frameworks, MUSCAT may not be directly comparable to other benchmark collections for the AMT and OMR fields. In this regard, we consider that the assortment would benefit from providing annotations for other transcription frameworks (*e.g.*, piano-roll representations as ground truth for AMT or symbol-level bounding boxes for OMR), at least for comparative purposes.

Regarding the MUSCUTS assortment, the main drawback of the dataset is that no taxonomy to group and catalog its samples in terms of their features (*e.g.*, samples with a fixed number of voices, instruments or playing difficulty, among others) is provided. Such a feature would allow posing controlled and well-defined experimental scenarios to extract meaningful insights and the study of particular training strategies—*e.g.*, curriculum learning—to further enhance the performance of score-level AMT schemes.

Finally, we also acknowledge that, while freely released for its use, MUSCAT and MUSCUTS will benefit from being integrated into tools and libraries commonly considered by the MIR community to simplify the development and reproduction of experimental pipelines. Particularly, we consider that integrating these assortments as part of the well-know *mirdata* library [5] will remarkably foster its use in the community.

## 4 COLLECTION ACCESS

The current collection versions —MUSCAT1.0 and MUSCUTS1.0— have been released for research-oriented purposes. Note that, as commonly done in the MIR community, these collections are described following the recommendation by [27] for the specification of MIR corpora.

Besides, to collaboratively increase the size of the collections, a web-based platform has been developed so that researchers may contribute to them. Moreover, this application provides user-friendly navigation across the data and allows for automatically computing summarizing statistics associated with the elements in the collection. Figure 7 shows some screenshots of this platform.

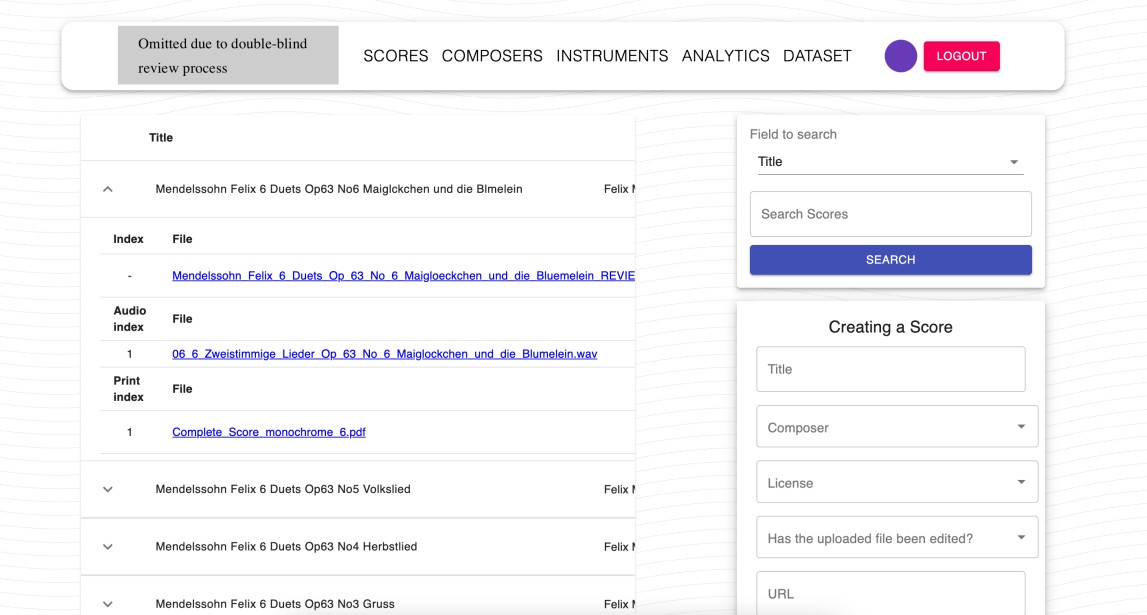

Fig. 7. Web service implemented for collaborative data collection.

Finally, the data resources—the MUSCAT1.0 and MUSCUTS1.0 assortments—as well as the web-based platform are available, for research-oriented purposes, at: *[omitted due to double-blind review process. An anonymous copy has been provided for this phase of the review process].* [5]

## 5 CONCLUSIONS AND FUTURE WORK

Music transcription is deemed as one of the key processes within the Music Information Retrieval (MIR) area. Under this generic definition, the MIR community depicts two different—yet related—research lines: (i) Automatic Music Transcription when addressing acoustic music signals; and (ii) Optical Music Recognition when the data source is a document. Due to the difficulty of defining a common transcription framework, these tasks have historically been addressed separately. Nonetheless, developing novel proposals that model both tasks as *sequence labeling* problems has fostered the proposal of a multimodal formulation in which the acoustic recording and the sheet image constitute the individual modalities of a common target representation, a digital music score. However, while proven to outperform single-modality recognition rates, this approach has been exclusively validated under controlled scenarios—monotimbral and monophonic synthetic data—, being hence necessary to validate these premises on real-world data.

To address this shortage in the literature, this work introduces the *Multimodal mUSic Collection for Automatic Transcription* (MUSCAT) collection of real audio recordings and image sheets together with notation-level digital scores. The assortment comprises nearly 80 hours of real recordings with different instrumental combinations—from piano to orchestral music—and polyphony degrees, as well as

1251 scanned sheets and 880 symbolic scores from 37 composers. The data has been manually compiled from different sources on the Internet and later curated to remove possible error sources. To assess some length-dependent models in the state of the art, a fragmented version has been created, MUSCUTS, by manually annotating time intervals from MUSCAT up until more than 20 hours of audio. Additionally, a benchmark using the state-of-the-art score-level music transcription is provided on these data. While MUSCAT is expected to act as a reference and benchmark dataset for multimodal transcription, it may also be considered raw data in other MIR tasks involving audio and/or score data.

Future work mainly contemplates using this collection to benchmark existing stand-alone and multimodal score-oriented AMT and OMR frameworks to provide and point out limitations in the fields. Besides, we also contemplate tackling the commented limitations and needs of the assortment, such as its integration in reference MIR libraries and toolboxes to facilitate its use by the community. Finally, we intend to extend the collection to other music traditions different from the contemplated Common Practice Period—*e.g.*, non-Western traditions or modern music styles and genres—that allow obtaining other insights when proposing and benchmarking novel transcription methods.

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
