# OpenReview forum: "MUSCAT: a Multimodal mUSic Collection for Automatic Transcription of real recordings and image scores"
_acmmm.org/ACMMM/2024/Conference — MM2024 Oral_

### Official Review · Reviewer_kXAk · 2024-05-24

**Rating:** 3
**Confidence:** 3

**Summary:**

The paper presents the Multimodal mUSic Collection for Automatic Transcription (MUSCAT), which is a novel dataset specifically designed for the task of multimodal audio-image music transcription. This dataset includes real recordings and image scores together with their score-level annotations in several notation formats. MUSCAT is significant due to its variety, comprising nearly 80 hours of real recordings with a diverse range of instrumentation and polyphony, 1251 scanned sheets, and 880 symbolic scores from 37 composers. This collection is especially designed to address the limitations of existing collections which are mostly monotimbral, monophonic, and synthesized, thereby restricting the applicability of multimodal music transcription systems in realistic settings.
Methodologically, the MUSCAT dataset was meticulously assembled and curated, employing manual data collection and verification to ensure accuracy and relevance of the data. This involved a musicologist manually gathering and assessing each piece of music to confirm that the recorded and graphical scores represented the same work. The audio and image data underwent a standardization process before being added to the collection. Additionally, a subset named MUSCUTS focuses specifically on acoustic data for score-level automatic music transcription (AMT), providing a resource aimed at fostering further research in this challenging area.
The paper also discusses the creation of a baseline experiment using the MUSCUTS subset, outlining the methodology for engaging with this new dataset. The aim is to enable a more refined evaluation of AMT and Optical Music Recognition (OMR) systems under more varied and realistic conditions than previously possible. This paper not only provides a valuable resource for advancing research in the field of music information retrieval but also sets a benchmark for future explorations into multimodal music transcription.

**Strengths:**

The introduction of the Multimodal mUSic Collection for Automatic Transcription (MUSCAT) significantly advances the field of music information retrieval by addressing a critical gap: the lack of large-scale, diverse datasets of real recordings paired with image scores and symbolic music notations. This novelty is crucial because previous datasets predominantly utilized synthetic or limited scope data that do not reflect the complexities found in real-world music transcription tasks. The availability of such a comprehensive and realistic dataset enables researchers to tackle more sophisticated problems in automatic music transcription (AMT) and optical music recognition (OMR), potentially leading to more robust and adaptable systems.
The methodology employed in the collection and curation of the MUSCAT dataset underscores its technical correctness and potential for reproducibility. The meticulous manual process, overseen by professional musicologists, ensures high data integrity. This precision in data curation is vital, as it directly influences the performance and reliability of AMT and OMR systems trained using this dataset. By detailing the standardization processes and the selection of notation formats, the paper sets a high standard for dataset preparation that can serve as a model for future research in the area.
Furthermore, the baseline experiments conducted with the MUSCUTS subset provide an initial evaluation framework for the dataset. These experiments not only validate the utility of MUSCAT but also highlight the challenges of multimodal music transcription in realistic settings. By establishing a benchmark, the paper encourages further research and development, promoting an iterative cycle of improvement in multimodal transcription technologies.
The paper’s clarity in articulating its aims, the dataset’s creation, and the implications of its findings make it an exemplary resource for the community. The dataset’s provision to tackle complex transcription tasks, combined with its potential to support a variety of other music-related research areas such as composer recognition and instrument identification, positions MUSCAT as a cornerstone resource that will likely influence the field significantly.

**Limitations:**

Although the dataset introduces a wide range of real recordings and notations, it’s not clear how well this diversity covers less common or non-Western musical genres. If the dataset primarily includes Western classical music, its applicability to other music genres, such as jazz, folk, or various non-Western traditions, might be limited. This could be a significant oversight, as the generalization of AMT and OMR systems across different musical traditions is a critical aspect of their practical utility.
The baseline experiments described utilize the dataset to highlight the challenges in multimodal music transcription but might not sufficiently demonstrate the effectiveness of existing transcription systems or the dataset itself in advancing the state of the art. If the experiments only reveal the difficulty of tasks without contributing substantial new insights or methodologies for overcoming these challenges, the paper may fall short in pushing the field forward.
While the paper details the data curation and baseline experiments, it might lack sufficient technical details about the data processing steps or the specific configurations of the transcription systems tested. This lack of detail can impede reproducibility and make it difficult for other researchers to build directly upon this work.
If the dataset requires extensive manual curation, its scalability might be questioned. Building similar datasets for broader or more varied applications could be resource-intensive and impractical. Moreover, if the dataset or the tools developed for its curation and processing are not made openly accessible, it limits the community’s ability to use these resources effectively.

**Suitability:**

3

---

### Official Review · Reviewer_7yiE · 2024-05-24

**Rating:** 6
**Confidence:** 3

**Summary:**

This paper introduces a novel multimodal dataset tailored for research in both optical music recognition and automatic music transcription within multi-track settings. The dataset comprises 1,251 scanned music sheets, 880 symbolic scores, and 80 hours of real recordings. Importantly, the dataset presents major compositional elements—optical music, real recordings, and symbolic annotations—in a parallel format, significantly enhancing the prospects for future multi-track music transcription studies that utilize both optical and audio sources. A preliminary experiment was conducted on the audio-to-score transcription task, indicating substantial potential for further advancements in this area.

**Strengths:**

- The paper includes a thorough survey and comparative analysis of existing multimodal datasets for symbolic music recognition, integrating both optical music and audio sources.
- The meticulously assembled dataset promises to be a valuable resource for advancing research in optical music recognition and audio-to-score transcription, as well as transcription that takes both audio and optical scores as input.
- The initial experimental results underscore a considerable space for improvement in audio-to-score transcription research.

**Limitations:**

**Major Issues**

The reviewer recognized no major issues with the paper.

**Minor Issues**
- Section 3.2 lacks clarity
    - Regarding the symbolic annotation process, what information was obtained during human annotation? (For example, does the annotations include detailed timings, such as the start and end times of each musical note?)
    - It remains unclear how the annotation procedures in the MUSCUTS component differ from those employed in the rest of the MUSCAT dataset.

**Suitability:**

3

---

### Official Review · Reviewer_rUWc · 2024-05-24

**Rating:** 1
**Confidence:** 2

**Summary:**

This work aims to advance multimodal image and audio music transcription, as well as unimodal transcription. It addresses the scarcity of real data-based multimodal music corpora by compiling a diverse dataset. This dataset includes nearly 80 hours of real recordings with varied instrumentation and polyphony, from piano solos to orchestral music, along with 1,251 scanned sheets and 880 symbolic scores from 37 composers. Baseline experiments highlight the importance of fostering research in this field using real recordings. Additionally, a web-based service is provided to facilitate collaborative expansion of the collection.

**Strengths:**

The paper discusses the development of a comprehensive collection of real audio recordings and image sheets, which are paired with notation-level digital scores and a fragmented subset. This collection is relevant to the topics covered by ACM Multimedia (ACM MM). The authors have introduced a web-based service designed to enhance the size of these collections through collaborative efforts, allowing users to contribute and expand the available resources.

**Limitations:**

The author's name and affiliation can be found in the "Relevance To Conference" tab. By not concealing their identities, this paper violates the double-blind policy of ACM MM.

**Suitability:**

3

---

### Official Review · Reviewer_oYpp · 2024-05-25

**Rating:** 5
**Confidence:** 3

**Summary:**

The submission presents a new and large dataset (MUSCAT) specifically devised for multimodal score-oriented transcription tasks. It includes a carefully musicologist-checked assortment of acoustic recordings, image sheets and their score-level annotations in various notation formats. A subset of the dataset (MUSCUTS) focused on automatic music transcription at the score level is also presented. The paper emphasizes the need for research using real recordings and introduces a web-based service for collaborative data expansion.

**Strengths:**

- A reliable and accurate multimodal collection of music data, that were checked by professional musicologists
- The fragmented dataset exclusively focused on acoustic data for score level automatic music transcription
- Important tool for Music Information Retrieval community
- Authors expressed interest in incorporating into the tools and libraries commonly considered by the MIR community to simplify the development and replication of experimental pipelines.

**Limitations:**

- The paper does not appear to be formatted correctly according to the ACM latex style.
- Other limitations are discussed by authors in Sect. 3.3

**Suitability:**

3

---

### Meta-Review · Area_Chair_2DyE · 2024-07-03

**Recommendation:** Accept (Oral)
**Confidence:** 4

**Metareview:**

This paper is a controversial case that resulted in a delay in the meta review because I had to discuss it with the SAC.

This paper includes a thorough survey and comparative analysis of existing multimodal datasets for symbolic music recognition, integrating both optical music and audio sources. The meticulously assembled dataset promises to be a valuable resource for advancing research in optical music recognition and audio-to-score transcription, as well as transcription that takes both audio and optical scores as input. The initial experimental results underscore a considerable space for improvement in audio-to-score transcription research. All reviewers recognized the technical merits of this paper. Despite some relatively minor technical issues of this paper, I would have no problem to recommend an accept if only technical merits were considered.

However, there are two major problems with this paper that could result in a straightforward reject. These include: 1) The author's name and affiliation are revealed in the "Relevance To Conference" tab visible to all reviewers. By not concealing their identities, this paper violates the double-blind policy of ACM MM, as pointed out by Reviewer rUWc; 2) The authors did not follow the ACM MM submission template, as pointed out by Reviewer oYpp. To follow the ACM MM rules, I assign a reject here, but request that this case needs to be discussed during the TPC meeting, which will make the final decision.

***TPC Addendum***
The TPC discussed the peculiar circumstances for this paper. It is a great technical contribution, relevant to the theme of multimedia processing, and would likely make a good addition to the program. The anonymization breach is a serious issue that we cannot take lightly. However, the TPC felt that the anonymization issue came up in a rather peculiar place (relevance to conference) that was added to the process this year, and not in the paper or supplementary material of the paper. This resulted in a relatively late discovery of the issue. The TPC would like to give authors the benefit of doubt. Without that penalty, the paper should be accepted.